# Evaluation of Diaphragmatic Ultrasound in Respiratory Functional Assessment in Patients with ALS

**DOI:** 10.3390/diagnostics15070884

**Published:** 2025-04-01

**Authors:** Miguel Iglesias, Juan Alejandro Cascón, Andrés Maimó, Antonio Albaladejo, Felipe Andreo, Ana Sánchez Fernández, María Maciá Palazón, Isabel Martínez González-Posada, Ruth García García, Rosa Cordovilla

**Affiliations:** 1Servicio de Neumología, Hospital Universitario de Salamanca, 37007 Salamanca, Spain; miglesiash@saludcastillayleon.es (M.I.); asanchezfe@saludcastillayleon.es (A.S.F.); rgarciaga@saludcastillayleon.es (R.G.G.); 2Servicio de Neumología, Hospital Universitario Central de Asturias, 33011 Oviedo, Spain; jacascon@saludcastillayleon.es (J.A.C.); imartinezgp@gmail.com (I.M.G.-P.); 3Servicio de Neumología, Hospital Universitario Son Llàtzer, 07198 Palma de Mallorca, Spain; amaimo@hsll.es (A.M.); mariaasuncion.macia@ssib.es (M.M.P.); 4Servicio de Neumología, Hospital de Manacor, 07500 Manacor, Spain; antonioa.davalos@gmail.com; 5Servicio de Neumología, Hospital Universitario Germans Trias i Pujol, 08916 Badalona, Spain; fcandreo.germanstrias@gencat.cat

**Keywords:** Amyotrophic Lateral Sclerosis, thoracic ultrasound, respiratory function test

## Abstract

**Background:** Diaphragmatic ultrasound emerges as a valuable non-invasive method for assessing diaphragm functionality in patients with amyotrophic lateral sclerosis (ALS). This study aimed to evaluate diaphragmatic ultrasound parameters in ALS, compare them with respiratory function tests, and determine whether they are associated with the need for non-invasive ventilation (NIV). **Methods:** This was a prospective, descriptive, and multicenter study across five centers, enrolling patients with recent diagnoses of ALS. At three-monthly visits, participants underwent both diaphragmatic ultrasound and pulmonary function testing. The following variables were analyzed: withdrawal from this study due to NIV or death, excursion, velocity, thickness, thickening fraction, and spirometric and respiratory muscle function values. **Results:** A total of 41 patients were included. A total of 24 (61.5%) patients left this study before the final year: 17 due to initiation of NIV, 4 due to clinical deterioration without NIV, and 3 due to death. Statistically significant moderate correlations were observed between diaphragmatic excursion and velocity and FVC and supine FVC (*p* < 0.001) and with MIP and the SNIP test (*p* < 0.05). No correlation was observed with thickening fraction. Additionally, lower baseline values in excursion were significantly associated with study withdrawal, along with reduced lung function (FVC, supine FVC, and MEP (*p* < 0.001). **Conclusions:** assessing diaphragmatic excursion by ultrasonography may serve as a useful tool for monitoring patients with ALS.

## 1. Introduction

Neuromuscular disorder (NMD) is a term that encompasses a heterogeneous group of nosological entities resulting from a primary or secondary alteration of skeletal muscle cells. Although rare, they represent a significant source of disability. NMDs can affect the neuromuscular system at various levels, from the spinal cord, peripheral nerves, or neuromuscular junction to the muscles themselves.

Over the last few decades there has been a shift in the natural history of NMDs, largely due to improvements in the diagnosis and treatment of their respiratory complications (the leading cause of mortality in these patients) [1]. Therefore, it follows that pulmonary function should be assessed in all of these patients, even in the absence of symptoms, and should be monitored at regular intervals during follow-up.

Depending on the characteristic chosen to classify NMDs fundamentally alters their groupings, for example genetics or anatomy. The classification with the most significant impact on the standardization of care protocols is the one based on their temporal profile, distinguishing between slowly progressive, such as Duchenne muscular dystrophy, and rapidly evolving, the most prominent of which is amyotrophic lateral sclerosis (ALS). Due to the crucial role played by pulmonologists in ALS patient care, most ALS units include at least one respiratory medicine specialist as part of their multidisciplinary care.

ALS is an unknown etiology neuromuscular disease characterized by the fast and progressive degeneration of upper and lower motor neurons, usually progressing rapidly over months, or occasionally a few years, with respiratory involvement determining the prognosis [2]. The incidence of sporadic ALS in Europe and the United States is estimated to be between 2.7 and 7.4 per 100,000 inhabitants per year. It is more common in males, with average survival ranging from 20 to 48 months [3].

In around 80% of cases, the first clinical sign is asymmetric limb weakness, with the majority of the resulting 20% debuting with bulbar impairment, such as dysarthria or dysphagia [4]. Only 1–3% will present with respiratory muscle weakness at onset [4,5].

The progressive weakness of respiratory muscles is the most important predictor of survival. Therefore, it follows that lung function tests should be performed in all patients with ALS and result progression should be closely monitored throughout follow-up. Spirometry (in both sitting and supine positions), muscle strength tests, and blood gas analysis are all included as part of an initial assessment [2]. Other commonly used tests are maximal inspiratory pressure (MIP) and sniff nasal inspiratory pressure (SNIP); however, they show significant inter- and intra-patient variability, limiting their utility for monitoring and predicting the need for non-invasive mechanical ventilation (NIV). A similar situation is true for the use of forced vital capacity (FVC). While it correlates with diaphragmatic involvement, FVC may not be a useful marker for the need to initiate NIV in a subset patient with nocturnal hypoventilation who concomitantly preserves FVC values [2]. Hypercapnia typically occurs in later stages and is not a reliable indicator in the early phase of the disease [3].

Ultrasound is a non-invasive and effective tool for the assessment of diaphragm mobility and function. Various studies have demonstrated the applicability of diaphragmatic ultrasound as a comparable evaluation tool to more costly imaging techniques [6,7,8]. Measurements that can be taken of the diaphragm include thickness, thickening fraction, excursion, and velocity [9].

Diaphragmatic atrophy, determined by a reduced thickness at end-expiration and a low thickening fraction, is of great interest in ALS through its correlation with pulmonary function [10,11] and therefore as a possible indicator for the need for NIV, which has been shown to increase survival for these patients [12], with some authors pleading for its early initiation [13]. Although the decision to implement external ventilatory support is a multifactorial one, FVC remains the most widely accepted parameter for monitoring respiratory function [14].

Therefore, diaphragmatic ultrasound could be a useful tool for respiratory muscle function assessment in patients with ALS during clinical follow-up, along with the implementation of non-invasive ventilation. Further research is needed, specifically a comparative analysis with the established measures in clinical guidelines.

The main objective of our study was to glean the ultrasound-derived diaphragmatic metrics of patients with ALS and directly compare them with respiratory function test measurements. Additionally, our aim was to analyze whether a potentially predictive relationship exists between these measurements and the need for NIV.

## 2. Materials and Methods

A prospective, descriptive, and multicenter study (EcoELA) was conducted. Approval was previously requested and accepted from the research ethics committees of each participating hospital (CEIM Code: PI 2019-04-289). All patients diagnosed with ALS were referred to the specialized NIV clinics at each center between June 2019 and June 2023 and were included in consecutive order. Written informed consent was obtained from all participants after verbal and written information was provided. The participating centers were as follows: Hospital Universitario de Salamanca (HUS), Hospital Universitario Central de Asturias (HUCA), Hospital Universitario Son Llàtzer (HUSLL), Hospital de Manacor (HMAN), and Hospital Universitario Germans Trias i Pujol (HUGTP), all part of the Spanish Public Healthcare System in various regions of the country. In all centers, a monographic NIV clinic is carried out by a multidisciplinary team consisting of a pulmonologist, a neurologist, and a nurse.

Patients included in this study met the following criteria: diagnosed with ALS, monographic NIV clinic referral, or in follow-up but not undergoing NIV treatment. Patients actively receiving NIV treatment, those unable to comply with testing procedures due to overall functional decline, and those who opted-out of this study were excluded.

Diaphragmatic ultrasound was performed on all patients, starting in the first visit to the NIV clinic. During their first appointment, echography, lung function tests, and a complete respiratory muscle function assessment were conducted in the aforementioned order on the same day. Subsequent visits were scheduled every three months. There were four different reasons for patients to exit this study: initiation of NIV, clinical worsening, voluntary withdrawal, or death. NIV treatment was initiated as needed, based on clinical and blood gas criteria, in accordance with guidelines from the American College of Chest Physicians (ACCP) [15], the European Respiratory Society and American Thoracic Society (ERS/ATS) [16], and the European Federation of the Neurological Societies (EFNS) [14].

Ultrasound examinations were performed in the interventional pulmonology unit of each center by a pulmonologist experienced in thoracic and diaphragmatic ultrasound. For the subcostal approach, only the right hemidiaphragm was assessed through the liver using a convex probe with frequencies between 3.5 and 5 MHz. Otherwise, a high-frequency linear probe with a frequency range of 7 and 10 MHz was used for the axillary approach, following the recommendations of the Spanish Society of Pulmonology and Thoracic Surgery (SEPAR) procedures manual [9].

Measurements were obtained with the patient in the supine position, with the head of the bed elevated to approximately 30 degrees, after maintaining this position for at least 15 min. Following patient positioning, assessments were initiated using the subcostal approach to identify the posterior portion of the right hemidiaphragm. Its visibility was verified in both B-mode and M-mode during slow deep inspirations before instructing the patient to perform three rapid deep inspirations, then selecting the measurement with the highest value.

Once this measurement was obtained, the axillary approach was employed, between the 8th and 10th intercostal spaces along the mid-axillary line, bilaterally. After identifying the diaphragm, the segment nearest to its costal insertion was selected during deep inspirations, ensuring the absence of pulmonary parenchymal interference. Subsequently, the rapid and deep inspiration maneuvers were then repeated to obtain the highest values from the hemithorax where these were the highest (Figure 1).

Pulmonary function tests were performed on the same day following diaphragmatic ultrasound. These included spirometry in both the supine and sitting positions, measurement of maximal inspiratory pressure (MIP), maximal expiratory pressure (MEP), SNIP test, and baseline arterial blood gas analysis. Reference values were derived from the 2012 European Respiratory Society (ERS) Task Force of the Global Lung Function Initiative [17].

Following the completion of both tests, the variables analyzed included age, sex, reason for study withdrawal (initiation of NIV, clinical deterioration, or death), diaphragmatic excursion during deep breathing, diaphragmatic thickness at end expiration (EndE), thickening fraction, FVC, supine FVC, MIP, MEP, and SNIP test.

The diaphragmatic thickening fraction was calculated as follows:$$Thickening\;fraction\;\left(\%\right)\;=\frac{endI-endE}{endE}\times 100\;(“endI”\;stands\;for\;the\;diaphragmatic\;thickness\;at\;the\;end\;of\;inspiration)$$

An initial sample size calculation was performed to achieve a diagnostic statistical power of 90% with a 95% confidence level (alpha error of 5%), determining that a minimum of 86 patients with ALS was required.

Statistical analysis was conducted using the SPSS^®^ version 20 software package. Correlations between variables were determined using Pearson’s correlation coefficient.

## 3. Results

A total of 61 patients who attended the monographic NIV clinics were initially evaluated, of whom, 41 met the study inclusion criteria (Table 1). Of these, 24 patients (58.5%) did not complete the one-year follow-up, exiting the study for various reasons as follows: 17 (70.8%) due to NIV initiation, 4 (16.7%) due to clinical deterioration without NIV initiation, and 3 (12.5%) due to death (Figure 2).

We conducted an analysis to assess potential correlations between diaphragmatic ultrasound parameters and pulmonary function variables (Table 2), along with their relationship with study withdrawal due to NIV initiation or death (Table 3). This correlation analysis between thoracic ultrasound and pulmonary function demonstrated that both excursion and velocity exhibited a statistically significant correlation with FVC and supine FVC (*p* < 0.01) (Figure 3), along with with MIP and the SNIP test (*p* < 0.05). No statistically significant correlations were observed for the remaining variables.

Focusing on study withdrawal due to NIV initiation or death, we analyzed correlations between lung function and diaphragmatic ultrasound parameters at two time points: the baseline visit (Table 4) and the last visit prior to study withdrawal. At baseline, lower pulmonary function values—including FVC, supine FVC, and MEP (*p* < 0.01)—along with reduced diaphragmatic excursion on ultrasound (*p* < 0.01), were significantly associated with study withdrawal (Figure 4).

Additionally, as an indirect marker of clinical deterioration, at the last visit prior to study withdrawal, a correlation was identified between FVC, supine FVC, MIP, MEP, SNIP test, and diaphragmatic excursion measured by ultrasound (Figure 5).

## 4. Discussion

Challenges in performing accurate pulmonary function maneuvers in some patients with ALS have prompted the search for alternative methods to assess diaphragmatic function without requiring substantial patient cooperation. Numerous studies have investigated the correlation between diaphragmatic ultrasound parameters and pulmonary function, with particularly promising results for the measurements of diaphragmatic thickness and thickening fraction [18,19,20]. For instance, Hiwatani et al. [21] (2013) reported a correlation between diaphragmatic thickness and FVC, although the study did not address clinical outcomes. It was the first but not the only study, because subsequent investigations [8,11,22] have similarly demonstrated significant associations between diaphragmatic thickening fraction and pulmonary function measures. Based on these findings, we anticipated comparable correlations in our series. However, our analysis did not reveal a significant relationship between diaphragmatic thickness or thickening fraction and lung function parameters (FVC, MIP, or MEP). We attribute this discrepancy to the relatively preserved clinical condition of our patients at diagnosis (mean FVC 83%) and the early initiation of NIV in our centers, facilitated by rigorous follow-up in specialized monographic clinics. As a result, NIV was frequently initiated based on symptoms and modest declines in pulmonary function, thereby preventing these patients from progressing to severe reductions in FVC or inspiratory and/or expiratory pressures.

Although diaphragmatic thickness and thickening fraction are well characterized, other ultrasound parameters—such as excursion and diaphragm mobility velocity—have not been extensively evaluated in patients with ALS. To our knowledge, only Boussuges et al. [23], in quite an old study from 2009, demonstrated a clear correlation between right hemidiaphragm excursion and pulmonary function tests, but the study was limited to a healthy population. Measurement of these variables requires positioning the patient in a supine position with the head of the bed elevated approximately 30 degrees, using an abdominal convex probe. We adopted this position in our study to perform a comprehensive diaphragmatic ultrasound, which also enabled the assessment of patients who could not be evaluated in a seated position. Consistent with Boussuges et al. [23], our results showed that diaphragmatic excursion during deep inspiration was correlated with FVC.

Regarding the predictive capacity of diaphragmatic ultrasound for clinical deterioration, we found that diaphragmatic excursion was the only variable significantly associated with study withdrawal due to NIV initiation or death. Only two prospective studies in the literature have analyzed this predictive capacity. The first one, that of Fantini et al. [24], evaluated the utility of a cut-off point for the ratio between expiratory and inspiratory diaphragmatic thickness (0.75) in predicting NIV initiation, while Spiliopoulos et al. [25] reported a strong predictive capacity for the thickening fraction, with an area under the ROC curve of 0.989. Our study aimed to identify a variable at diagnosis that could predict which patients were most likely to require NIV and thereby serve as an early indicator of respiratory failure. Among our patients, those who experienced clinical deterioration showed significantly lower diaphragmatic excursion values both at baseline and at the last visit prior to study withdrawal, suggesting that a marked decline in diaphragmatic excursion may warrant NIV evaluation or, at least, closer monitoring.

Our study has several limitations. The main one is the variability in patient follow-up and the significant loss of visits due to the COVID-19 global pandemic, which complicated scheduling. Furthermore, because all procedures and patient management were carried out during routine clinical practice, not all tests could be completed at every visit, depending on individual patient characteristics. Additionally, although each center only had one or two examiners, there was notable inter-observer variability in thoracic ultrasound assessments, along with variability between ultrasound devices—a challenge commonly encountered in similar studies. Finally, our study is limited by its relatively small sample size, reflecting the low prevalence of ALS, and indicated the need for a larger more complex cohort study, with more emphasis on the disease onset.

## 5. Conclusions

Assessing diaphragmatic excursion during deep inspiration via ultrasonography may serve as a valuable tool adjunct for monitoring patients with ALS, particularly those unable to accurately perform FVC maneuvers or maintain a seated position. Further studies with larger cohorts are warranted to validate diaphragmatic excursion as a predictive measure for NIV initiation in patients with ALS.

## Figures and Tables

**Figure 1 diagnostics-15-00884-f001:**
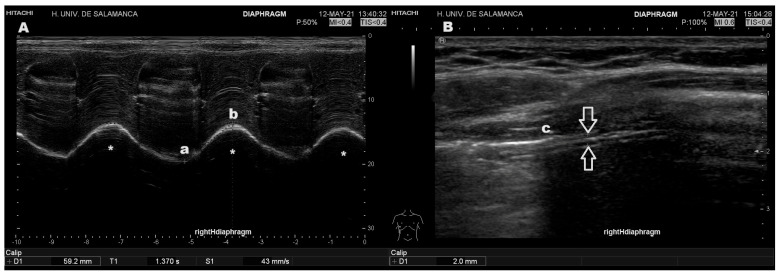
Typical ultrasound images of diaphragmatic ultrasound. (**A**) Measurement of excursion in M-mode, demonstrating waveforms corresponding to three deep inspirations with waves of three deep inspirations (*). Excursion is measured as the distance between the lowest point at end expiration (a) and the highest point (b) following deep inspiration. Velocity is the relation between that distance and the time to reach the highest point. (**B**) Measurement of diaphragmatic thickness in B-mode (the diaphragm showed as a sliding three-layer structure between the white arrows), obtained at end inspiration immediately below the inferior lung border (c).

**Figure 2 diagnostics-15-00884-f002:**
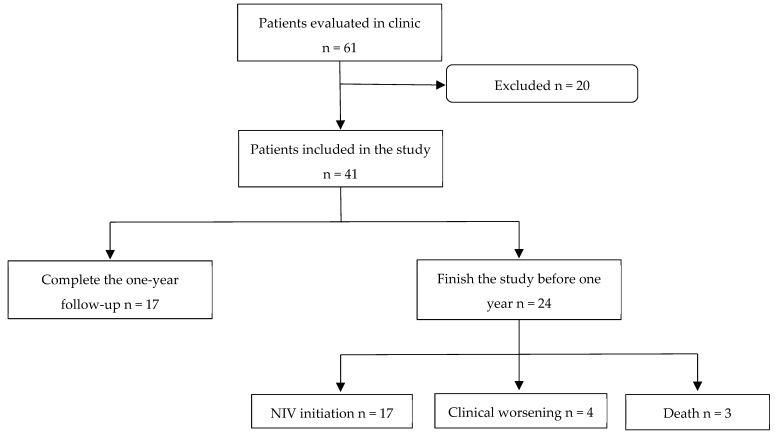
Patient flow diagram of this study.

**Figure 3 diagnostics-15-00884-f003:**
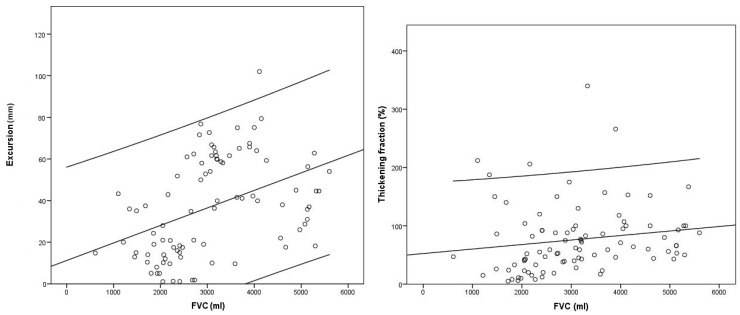
Correlation diagram between diaphragmatic excursion (**left**) and thickening fraction (**right**) and FVC.

**Figure 4 diagnostics-15-00884-f004:**
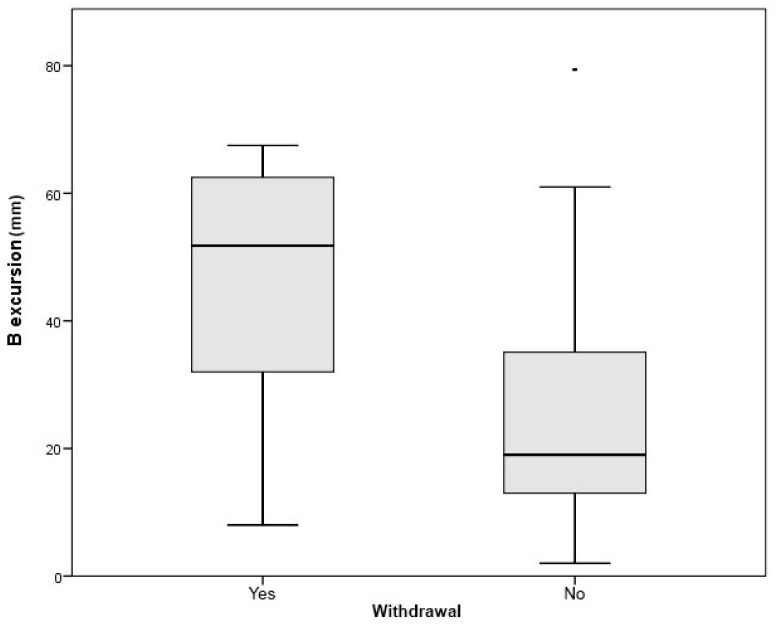
Box plot showing the significant difference in excursion values at the first visit (B excursion) between the group of patients who did not withdraw from this study (Withdrawal) and those who did.

**Figure 5 diagnostics-15-00884-f005:**
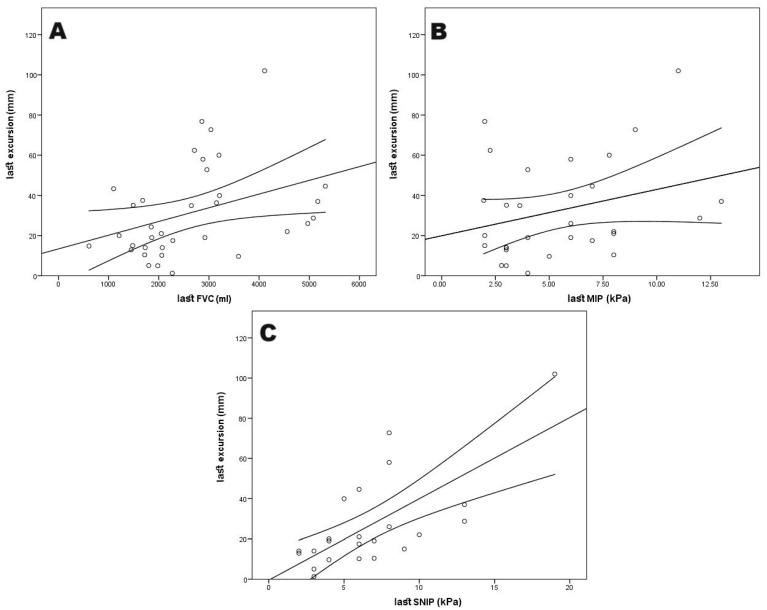
Correlation diagram of the last visit values between diaphragmatic excursion (last excursion) and (**A**) FVC (last FVC); (**B**) MIP (last MIP) and (**C**) SNIP test (last SNIP).

**Table 1 diagnostics-15-00884-t001:** Demographic characteristics of the study population (n = 41).

Mean age (years)	63.8
Sex (female)	21 (51.2%)
Year of diagnosis	
2017	3 (7.3%)
2018	4 (9.8%)
2019	3 (7.3%)
2020	14 (34.2%)
2021	6 (14.6%)
2022	6 (14.6%)
2023	5 (12.2%)
Participating center	
HUS	10 (24.4%)
HUCA	13 (31.7%)
HUSLL	12 (29.3%)
HMAN	4 (9.8%)
HUGTP	2 (4.9%)

**Table 2 diagnostics-15-00884-t002:** Correlation between diaphragm ultrasound and pulmonary function tests.

	DIAPHRAGM ULTRASOUND					
	Excursion		Velocity		Thickening Fraction	
	Pearson	*p*	Pearson	*p*	Pearson	*p*
FVC	0.420	0.000	0.339	0.008	0.156	0.100
FVC (%)	0.334	0.001	0.307	0.017	0.092	0.387
Supine FVC	0.466	0.000	0.343	0.007	0.320	0.086
Supine FVC (%)	0.353	0.002	0.226	0.083	0.209	0.067
MIP	0.225	0.040	0.273	0.040	0.018	0.873
SNIP test	0.319	0.017	0.432	0.003	0.091	0.504

**Table 3 diagnostics-15-00884-t003:** Correlation between diaphragm ultrasound values and withdrawal from this study.

	DIAPHRAGM ULTRASOUND			
	Baseline Excursion		Last Visit Excursion	
	Pearson	*p*	Pearson	*p*
Study Withdrawal	−0.458	0.004	−0.427	0.008

**Table 4 diagnostics-15-00884-t004:** Mean values at the baseline visit.

DIAPHRAGMATIC ULTRASOUND	
Excursion (mm)	33.0 (±21.4)
Velocity (mm/s)	13.2 (±8.3)
EndE thickness (mm)	1.9 (±1)
Thickening fraction (%)	75.5 (±59.6)
PULMONARY FUNCTION	
FVC (mL)	2966.7 (±1264.3)
FVC (%)	83 (±28.3)
Supine FVC (mL)	2910.1 (±1214.2)
Supine FVC (%)	82 (±25.4)
MIP (kPa)	5.5 (±2.5)
MEP (kPa)	6.9 (±3.6)
Snip test (kPa)	5.9 (±2.5)
ARTERIAL BLOOD GAS ANALYSIS	
pH	7.43 (±0.02)
pCO_2_ (mmHg)	40.3 (±7.6)
pO_2_ (mmHg)	83.8 (±11.7)
HCO_3_ (mmol/dL)	25.5 (±3.6)
Base excess	2.1 (±2.7)

## Data Availability

The research data are unavailable due to privacy and ethical restrictions.

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
