# Peer review of "Evaluation of Diaphragmatic Ultrasound in Respiratory Functional Assessment in Patients with ALS"

_diagnostics, 2025, doi:10.3390/diagnostics15070884_

Round 1
Reviewer 1 Report
Comments and Suggestions for Authors
Although there have been several reports of diaphragmatic ultrasound for ALS, I think the originality of this study is two-fold: first, it was a multicenter, prospective, multiple examination to examine the course of the disease, second, it describes the comparison with respiratory parameters other than FVC, i.e., SNIP and MIP/MEP.
The most striking, and at the same time problematic, aspect of this paper is its use of diaphragm excursion as the most useful parameter. This is because this parameter is relatively uncorrelated with other respiratory parameters and has been reported to be less useful in many cases.
eg.
- Aliberti S, et al. ALSFTD 14:154, 2013
- Morishima R, et al. Neurological Sciences 43:6821, 2022
In this paper, the evaluation of thickness in the zone of apposition of diaphragm is not as important, but in the context of previous papers, the emphasis has been more on that side. This divergence from the previous context should be given due consideration.
One concern that now accompanies this is concentrated in the diaphragm in figure 1B that is presented. The diaphragm shown in this figure is very uneven. In my own experience, the diaphragm in this position does not have this shape and is depicted as nearly parallel, though not perfectly. Of course, the area between the two high-intensity lines is the diaphragm, but I cannot dispel the suspicion that the diaphragm evaluated here is really a marginal site, one that actually exerts very little contractile force, and that this is why the correlation is weak.
Another concern about the study design is that it includes little assessment of the disease or the speed of progression of ALS itself. At the very least, height, weight, BMI, and duration from disease onset should be included. Also, information on whether the patient was bulbar onset is needed, as this will affect the measurement of respiratory parameters. If possible, the ALSFRS-R should be included in papers dealing with ALS.
Finally, the actual measured FVC is often used for statistics; shouldn't %FVC be used for this?
MINOR POINTS
The abbreviations in the figures, especially for the axes, are not clear: CVF in Figure 3, Salidaest and B excursion in Figure 4, and U excursion, U PIM, and U SNIP in Figure 5, which seem to be coined words. They seem to be coined terms, and should be changed to more easily understandable terms in the figures.
Figure 4, what is EXIT should be clearly stated in figure legend.
Missing graphs for data listed as correlated: supine FVC in Figure 3, FVC, supine FVC, MEP in Figure 4, FVC, supine FVC, MIP, MEP, SNIP in Figure 5. These may not be echo data though, These may not be echo data, but they need to be shown in some way.
Comments on the Quality of English Language
none
Author Response
Comment 1: In this paper, the evaluation of thickness in the zone of apposition of diaphragm is not as important, but in the context of previous papers, the emphasis has been more on that side. This divergence from the previous context should be given due consideration.
Response 1: Obviously, after reviewing the literature, the study was initially focused on finding a clinically useful relationship between diaphragmatic thickness measurements. However, we added to the ultrasound procedures the measurements of excursion and velocity. It was only later, during the statistical analysis, that we discovered the relationship with excursion and the lack of statistical significance with the diaphragm thickening fraction. In the discussion section of the article (lines 224 to 232), it is mentioned that both diaphragm thickness and diaphragm thickening fraction have already been properly validated for assessing its function using ultrasonography. We looked for reaching the same conclusion in our study, but unfortunately, we could not, probably due to the study's limitations. We have expanded the explanation of this topic in the discussion, hoping it could help to understand our job.
Comment 2: One concern that now accompanies this is concentrated in the diaphragm in figure 1B that is presented. The diaphragm shown in this figure is very uneven. In my own experience, the diaphragm in this position does not have this shape and is depicted as nearly parallel, though not perfectly. Of course, the area between the two high-intensity lines is the diaphragm, but I cannot dispel the suspicion that the diaphragm evaluated here is really a marginal site, one that actually exerts very little contractile force, and that this is why the correlation is weak.
Response 2: The procedure used for measuring diaphragmatic thickness has routinely followed the recommendations of guidelines and regulations: a linear probe placed in the anterior axillary line at the 8th and 10th intercostal spaces, aiming for the most marginal insertion possible. The image selected to illustrate the article corresponds to a healthy patient, which explains its dimensions, with the aim of graphically representing the measurement.
Comment 3: Another concern about the study design is that it includes little assessment of the disease or the speed of progression of ALS itself. At the very least, height, weight, BMI, and duration from disease onset should be included. Also, information on whether the patient was bulbar onset is needed, as this will affect the measurement of respiratory parameters. If possible, the ALSFRS-R should be included in papers dealing with ALS.
Response 3: The study design was carried out by pulmonologists with the aim of finding a prognostic method for assessing the need for non-invasive mechanical ventilation. Patient recruitment was conducted through referrals to a specialized clinic focused on initiating non-invasive mechanical ventilation. As a result, in many cases, there is no information on the onset of symptoms, nor on the application of prognostic scales for ALS, such as the ALSFRS-R. Nonetheless, thank you—this is a valuable point to consider for implementation in future studies.
Comment 4: The abbreviations in the figures, especially for the axes, are not clear: CVF in Figure 3, Salidaest and B excursion in Figure 4, and U excursion, U PIM, and U SNIP in Figure 5, which seem to be coined words. They seem to be coined terms, and should be changed to more easily understandable terms in the figures.
Response 4: Agreed, we'll fix it. The variables in the graphs are those used in Spanish databases, and we did not take this into account in the translation.
Comment 5: Figure 4, what is EXIT should be clearly stated in figure legend.
Response 5: We´ll fix it.
Comment 6: Missing graphs for data listed as correlated: supine FVC in Figure 3, FVC, supine FVC, MEP in Figure 4, FVC, supine FVC, MIP, MEP, SNIP in Figure 5. These may not be echo data though, These may not be echo data, but they need to be shown in some way.
Response 6: The relationship between pulmonary function measurements and the need for non-invasive mechanical ventilation has been previously demonstrated, and we do not believe it requires more information beyond mentioning it to avoid taking it for granted.
Reviewer 2 Report
Comments and Suggestions for Authors
Evaluating the utility of diaphragmatic ultrasound assessment in patients with ALS is a useful endeavor and represents a gap in the medical literature. I believe the investigators did a good job collecting ultrasound data, PFTs and patient data. They report finding significant correlations, but it is not clear what statical methods they used. I did not see an R value reported. The correlation looks somewhat weak based on the graphs, however the graphs are not well labeled and there for a difficult to interpret. Parts of the article are well written, while other parts need significant revision based on grammar and sentence syntax. I think with significant revision including better description of statical methods, better labeling of graphs, this paper would me much improved .
Points for clarification or revision
· Its not clear what statistical analysis was done to assess for correlation between diaphragmatic ultrasound findings and PFTs (I did not see an R value (pearson correlation reported).
· Table 1: not clear that the year of diagnosis is relevant information, I would probably cut it out of this table
· Figure 3: Unites on the y-axis are not listed. Why is Excusion 80,000 on one graph and 20—60 on another? Is this CM, MM???
· The Text reports that diaphragm velocity was measure, but not described how. Was tissue doppler used? No rate reported.
· In figure 3, the x-axis is labeled CVF but the caption says FVC. Is this a typographical error?
· In figure 4, the x-axis is labeled with a work not in the English language. The boxes are not labeled, so unclear which represents the group that left the study
· The investigators report a significant difference in baseline excuse between those who left the study for NIV and those who did not, but the box plots appear to overlap. Are the whiskers standard error, or total value? Would using the median rather than mean separate the boxes.
· Can table 2 but duplicated but have two columns (one for those that exited the study and one for those that did not
· The discussion needs significant editing/ revision based on grammar and word choice. If not done, would recommend having it reviewed by a native English speaker. At times incorrect word choices were used (for example line 229, mentions “allowed us to explore patients.” Explore is not the best word choice.
· It would be good to have a better description of what a “monographic NIV clinic” is.
· Sentence on line 106 and 107 has very awkward syntax, recommend revision
“In the first visit, and on the same day, firstly was performed the ultra-106 sonography examination and, immediately after, the complete pulmonary and respira-107 tory muscle function tests” Please revise.
Comments on the Quality of English Language
See other section. I would recommend having it reviewed by a native English speaker. At times word choice was incorrect or at best sub-optimal. The discussion appeared to be the section that would require the most revision.
Author Response
Comment 1: Its not clear what statistical analysis was done to assess for correlation between diaphragmatic ultrasound findings and PFTs (I did not see an R value (pearson correlation reported).
Response 1: We will add two tables with the R values to the article
Comment 2: Table 1: not clear that the year of diagnosis is relevant information, I would probably cut it out of this table
Response 2: The duration of disease progression from diagnosis to assessment by a pulmonologist is relevant as a determinant of the level of functional impairment at the time of recruitment. That’s why that information may appear at least as orientation.
Comment 3: Figure 3: Unites on the y-axis are not listed. Why is Excusion 80,000 on one graph and 20—60 on another? Is this CM, MM???
Response 3: Agreed, we’ll fix it
Comment 4: The Text reports that diaphragm velocity was measure, but not described how. Was tissue doppler used? No rate reported.
Response 4: The velocity measurement is performed, as recommended by guidelines and literature, using the analysis of the curve obtained with the subcostal approach, employing a convex probe and M-mode. We will add an explanatory note in Figure 1.
Comment 5: In figure 3, the x-axis is labeled CVF but the caption says FVC. Is this a typographical error?
Response 5: Yes, it’s a translation mistake. We’ll fix it
Comment 6: In figure 4, the x-axis is labeled with a work not in the English language. The boxes are not labeled, so unclear which represents the group that left the study
Response 6: Agreed, we’ll fix it
Comment 7: The investigators report a significant difference in baseline excuse between those who left the study for NIV and those who did not, but the box plots appear to overlap. Are the whiskers standard error, or total value? Would using the median rather than mean separate the boxes.
Response 7: The quartiles in the box plot overlap, but this does not imply that there is no statistical significance. As for the whiskers, they represent the total values, not the standard deviation.
Comment 8: Can table 2 but duplicated but have two columns (one for those that exited the study and one for those that did not
Response 8: Agree, if that data may make the results clearer, we’ll add that columns
Comment 9: The discussion needs significant editing/ revision based on grammar and word choice. If not done, would recommend having it reviewed by a native English speaker. At times incorrect word choices were used (for example line 229, mentions “allowed us to explore patients.” Explore is not the best word choice.
Response 9: The new version of the article has been reviewed by a native English speaker to correct syntax and writing issues.
Comment 10: It would be good to have a better description of what a “monographic NIV clinic” is.
Response 10: Agreed, we’ll add more information in the new version
Comment 11: Sentence on line 106 and 107 has very awkward syntax, recommend revision
Response 11: We will revise that with a native English speaker
Round 2
Reviewer 1 Report
Comments and Suggestions for Authors
I have checked the corrections to the areas you have pointed out.
However, there are still concerns about the quality of the various locations.
- Regarding Figure 1B, I received your reply that you followed the method described. However, honestly speaking, I do not believe that what is depicted as the diaphragm in this figure is the diaphragm. I would like to see a figure that more clearly resembles THE DIAPHRAGM. Otherwise, the measurement of the thickness of the diaphragm is in serious uncertainty.
- Regarding figure5, what is the U in front of FVC, MIP, SNIP?
Author Response
Comment 1: Regarding Figure 1B, I received your reply that you followed the method described. However, honestly speaking, I do not believe that what is depicted as the diaphragm in this figure is the diaphragm. I would like to see a figure that more clearly resembles THE DIAPHRAGM. Otherwise, the measurement of the thickness of the diaphragm is in serious uncertainty.
Response 1: We've changed the figure with another one that, actually, can show better the diaphragm as we are more used to see it, with the three lines and the curtain sign by the interposition of the expanded lung.
Comment 2: Regarding figure5, what is the U in front of FVC, MIP, SNIP?
Response 2: In order to ​​identify the values ​​of each one of the variables across the study, from the basal visit until the last visit before the study sale, we identify the values ​​of this last one with the letter U (from the spanish word for last: ULTIMA). with the idea of ​​identifying the values ​​of each one of the variables across the study, from the basal visit until the last visit before the study sale, we identify the values ​​of this last one with the letter U. The objective of not removing it from the graphics was to avoid confusion with the values ​​at another point in the continuation of the study. If taking those "U" away does improve the understanding, we won't have any problem with the removal.
Reviewer 2 Report
Comments and Suggestions for Authors
The article seems much improved from the prior version . Listing the pearson coefficient was helpful. I would probably list it in the abstract as well for the FVC and supine FVC. This value can be described as a moderate correlation.
I think it is worth mentioning in the abstract that diaphragm thickening did not correlated with FVC. In table 2 I would consider adding a third column for diaphragm thickening to show what that coefficient is. Similarly showing the negative data (diaphragm thickening vs FVC) in figure 4 would highlight the differences. Also in figure 4, I am not sure what the letter U is before all the labels on both the X and Y axis. Does it stand for something?
Author Response
Comment 1: Listing the pearson coefficient was helpful. I would probably list it in the abstract as well for the FVC and supine FVC. This value can be described as a moderate correlation. I think it is worth mentioning in the abstract that diaphragm thickening did not correlated with FVC.
Response 1: I've added the moderate term to the abstract and mentioned the lack of correlation with the thickening fraction, but with just 200 words, and the other modifications suggested, It has been impossible to add the Pearson correlation to it
Comment 2: In table 2 I would consider adding a third column for diaphragm thickening to show what that coefficient is.
Response 2: I've added that third column
Comment 3: Similarly showing the negative data (diaphragm thickening vs FVC) in figure 4 would highlight the differences
Response 3: That data should be better shown in Figure 3, so I've added it too
Comment 4: Also in figure 4, I am not sure what the letter U is before all the labels on both the X and Y axis. Does it stand for something?
Response 4: Correct, I've changed the variables with that "U" to "last"
Round 3
Reviewer 1 Report
Comments and Suggestions for Authors
As for Figure 1, it has improved and is barely recognizable as a diaphragm, but usually the diaphragm is a bit clearer, even in ALS patients. If you are going to publish a paper, it would be desirable to include a better figure.
This is an English-language paper, not a Spanish-language paper, and should be written in a manner that is discernible to non-Spanish speakers. In other words, it is not appropriate to use U to mean Ultima.
It appears that you have not responded to my comments regarding references. I will restate them;
The authors state that the speed of diaphragm movement has never been studied in ALS patients, however, did you checked the paper of Morishima R, et al. Neurological Sciences 2022 [PMID 36042062]? This paper states that there is a difference in FVC, but that velocity did not correlate with FVC.
Author Response
Comment 1: As for Figure 1, it has improved and is barely recognizable as a diaphragm, but usually the diaphragm is a bit clearer, even in ALS patients. If you are going to publish a paper, it would be desirable to include a better figure.
Response 1: I've added two arrows to figure 1, showing the diaphragm, and a little amendment to the explanation. I've attached a video (as non-published material) with an examination performed today on a patient with very serious neuromuscular involvement, to demonstrate the procedure.
Comment 2: This is an English-language paper, not a Spanish-language paper, and should be written in a manner that is discernible to non-Spanish speakers. In other words, it is not appropriate to use U to mean Ultima.
Response 2: Right, I've changed the variable with that "U" to "last", more appropiate.
Comment 3: The authors state that the speed of diaphragm movement has never been studied in ALS patients, however, did you checked the paper of Morishima R, et al. Neurological Sciences 2022 [PMID 36042062]? This paper states that there is a difference in FVC, but that velocity did not correlate with FVC.
Response 3: I'm sorry, I didn't notice it on my first review. I didn't review that article and don't currently have access to the full text (I've requested it from my institution). But it seems, by the abstract, that they weren't, actually, looking for that correlation. I'll correct it as soon I can read it properly.
Round 4
Reviewer 1 Report
Comments and Suggestions for Authors
I've read the revised article and now the image has improved sufficiently for publication.
Although there are some unsatisfactory points, such as insufficient literature review, I leave the final decision to the Editor